

# Modes of Vertical Thermodynamic and Wind Variability over the Maritime Continent

Jennie Bukowski[1], Derek J. Posselt[1], Jeffrey S. Reid[2], Samuel A. Atwood[3]

[1]University of Michigan, Ann Arbor, MI, USA
[2]Marine Meteorology Division, Naval Research Laboratory, Monterey, CA, USA
[3]Department of Atmospheric Science, Colorado State University, Fort Collins, CO, USA

*Correspondence to*: Derek J. Posselt (Derek.Posselt@jpl.nasa.gov)

**Abstract.** The Maritime Content (MC) is an exceedingly complex region, both from the perspective of its meteorology and
its aerosol characteristics. Convection in the MC is ubiquitous, and assumes a wide variety of forms under the influence of
an evolving large scale dynamic and thermodynamic context. Understanding the interaction between convective systems and
their environment, both individually and in the aggregate, requires knowledge of the dominant patterns of spatial and
temporal variability. To this end, radiosonde observations from 2008-2016 are examined from three sounding release sites
within the MC for the purpose of exploring the dominant vertical temperature, humidity, and wind structures in the region.
Principal Component Analysis is applied to the vertical atmospheric column to transform patterns present in radiosonde data
into canonical thermodynamic and wind profiles for the MC. Both rotated and non-rotated principal components are
considered, and the emerging structure functions reflect the fundamental vertical modes of short-term tropical variability.
The results indicate that while there is tremendous spatial and temporal variability across the MC, the primary modes of
vertical thermodynamic and wind variability in the region can be represented in a lower-dimensional subspace. In addition,
the vertical structures are very similar among different sites around the region, though different structures may manifest
more strongly at one location than another. The results indicate that, while very different meteorology may be found in
various parts of the MC at any given time, the processes themselves are remarkably consistent. The ability to represent this
variability using a limited number of structure functions facilitates analysis of co-variability between atmospheric structure
and convective systems, and also enables future systematic model-based ensemble analysis of cloud development,
convection, and precipitation over the MC.

## 1 Introduction

The Maritime Continent (MC) constitutes a region in the Western Pacific critical to the formation and/or evolution of many
of the key drivers of tropical atmospheric variability (Neale and Slingo, 2003). It plays host to a broad spectrum of
convective systems that interact with the environment and each other in ways that are difficult both to observe and to model.
It is also the site of one of the most complex aerosol environments on Earth (Reid et al., 2012; Atwood et al. 2013; Reid et





al., 2015; 2016a; 2016b; Atwood et al. 2016). Despite its significance to the global climate system, the MC is an area in which models struggle to reproduce the full range of observed atmospheric fluctuations (Jourdain et al., 2013). At the heart of the challenge is the fact that short- to medium-term variability in the region is affected by processes that span a tremendous range of temporal and spatial scales (Zhang, 2014). Among these are the diurnal cycle of convection, the

Madden-Julian Oscillation (MJO; Zhang, 2005), equatorial waves, seasonal monsoons (Misra and Li, 2014) progression of the Intertropical Convergence Zone (ITCZ), the El Niño Southern Oscillation (ENSO), as well as a host of mesoscale and local circulations. Models have difficulty realistically simulating the propagation of tropical waves through the region in part due to the interactions across scales (Peatman et al. 2014; 2015). They have also proven largely unable to reproduce the diurnal cycle of convection, which exhibits a nocturnal maximum over the ocean (Chen and Houze, 1997; Sui et al., 1997;

Yang and Slingo, 2001; Dai, 2001). Taken together, the spectrum of convective features observed in the MC are non-independent, their interactions are non-linear, and many are difficult to capture in regional and global models. All of them are sensitive to, and feedback on, the tropical vertical thermodynamic and wind structure, which serve as both a driver and consequence of the state of discernable weather and climate.

Several previous studies have investigated the dominant modes of tropical, and specifically, Maritime Continent climate

variability, in terms of rainfall (Teo et al., 2011; Yanto et al., 2016; Wang et al., 2014) and weather types (Moron, et al., 2015). Thermodynamic variability in the tropics has also been approached from a spatial framework (Serra et al. 2014), and a number of studies have used sounding datasets to examine tropical vertical thermodynamic and wind signatures (Brown and Zhang, 1996; Folkins et al. 2008) and diabatic heating profiles (e.g., Zhang and Hagos, 2009). While these studies have provided insight into the mean thermodynamic structure of the tropics and the range of variability, none have yet focused on

the MC specifically, nor have there been attempts to represent tropical troposphere variability in terms of vertical patterns that summarize the dominant modes of variability in the system. These vertical modes contain information on the system itself, and can also be used in numerical model-based analysis of tropical systems.

Specifically, consider a model-based analysis of the response of convective systems to changes in the environment, which would require systematic perturbation of the various factors that control the outcomes of convection (e.g., precipitation,

latent heating, radiative fluxes). The input factors should include at minimum the details of the vertical thermodynamic structure (e.g., temperature and water vapor content) that determine buoyancy (and hence convective vertical motion and detrainment) as well as the organizing influence of wind shear (changes in the u and v components of the wind with height). As a thought experiment, let us assume that a particular environment can be characterized by a representative sounding consisting of temperature, relative humidity, u, and v; and let us further assume that this sounding is resolved in 25 hPa

intervals from the surface (1000 hPa) to the lower stratosphere (50 hPa). If one were to examine the sensitivity of convection to changes in this sounding via brute force, one would need to perturb each variable of interest at each level independently. If we consider a simple experiment in which each variable at each level is perturbed just once, independent of the others, this equates to (4 variables x 39 layers)$^2$ = 24,336 simulations. If we consider a broader range of variability so that we require more than two values of each thermodynamic and wind variable at each level, this becomes (4 variables x 39 layers)$^N$, where



N is the number of discrete values of each variable to be examined at each level. This is clearly computationally unfeasible, and probably non-sensical since it is clear from observations that there is co-variability between different layers, and that there are large-scale patterns of variability that should allow reduction in the number of effective degrees of freedom in the system. The fundamental questions, therefore, are:

1.  Can the modes of vertical and temporal variability in MC soundings be reduced to a smaller set of coherent structures?

2.  How do these modes of variability change among different sub-regions in the MC, including those that are located in the remote ocean, those that are near significant topography, and those that are proximal to larger land masses?

This study answers these questions by utilizing soundings from three sites in the Maritime Continent region spanning a

period of eight years, in concert with an Empirical Orthogonal Function/Principal Component analysis, to explore the dominant modes of vertical variability in the MC. Each site is representative of a different MC sub-region, and the results are examined to explore the differences in modes obtained across the region. The results are useful not only for understanding the local meteorological variability, but also for informing future model-based sensitivity experiments. The remainder of this manuscript is organized as follows. Data sources are described in Section 2, while the analysis methodology is presented in

Section 3. Results of the EOF/PC analysis are presented in Section 4, while a summary and conclusions are offered in Section 5.

## 2 Data Sources

Radiosonde observations from 2008-2016 were collected from three release sites, including Ranai, Indonesia (Riau Island), Puerto Princesa, Philippines, and Singapore (Fig. 1) from 0Z and 12Z soundings. These three sites were selected to represent

the spatial extent of the MC and because of their reliable radiosonde release record. Since the Ranai sounding site has a unique location representative of MC meteorology, it will be featured as a case study in Section 4a and then compared with the other two sites in Section 4b. The specific radiosonde equipment, accuracy, and observational biases from each MC site were documented in Ciesielski et al. (2014). Because the instrument error is consistent among the three sites (Ciesielski 2016, pers. comm.), any biases that exist in the radiosonde data will not propagate into the EOF/PC analysis. Constant-bias

corrections are available for the three sounding sites but are not vital to the study and were not applied.

Raw vertical temperature (T), relative humidity (RH), and wind data (U and V) was obtained and decoded from the University of Wyoming atmospheric sounding database. Using linear-in-log-pressure interpolation, each variable was mapped from its original pressure levels to 10 hPa increments from 1000 hPa to 50 hPa for a total of 96 vertical levels. To be considered in the analysis, data points were required to report through the 40 hPa level, as no extrapolation techniques were

employed. While the focus of this study is the troposphere, the 50 hPa cutoff above, rather than at the tropical tropopause was deliberate: the transition from the troposphere to the stratosphere is an important signal in the system and should be included. Additionally, the PCA algorithm can be skewed if the end points are located at a level where there is a strong





signal. As such, levels above the tropopause were incorporated as a buffer so that the entire signal may be effectively captured. To increase the number of data points, each sounding variable was reviewed separately; retention of any one of T, RH, U, and V did not depend on all others being present through 40 hPa. However, because the U and V wind components were derived from the same values, if only one of the components failed the QC tests they were both rejected. As a final

step, a five point boxcar moving average filter was applied once to the vertical profiles to smooth the sounding curves. Quality control (QC) was then performed on the set of sounding variables. Because PCA is highly sensitive to outliers and bad data points, the soundings were rigorously inspected for reporting, decoding, and interpolation errors. Various checks were applied to each variable based on the climatology of the region in order to remove overtly erroneous soundings.

In addition to the objective quality control criteria listed in Table 1, the database was hand-analyzed twice during the QC

process. First, in order to remove poor logarithmic interpolations resulting from extended gaps in reporting levels, the maximum and minimum data points for each 10 hPa level were examined and vertical profiles with extreme values were discarded. Second, every Skew-T / Log-P diagram of the raw data sounding was visually inspected (Ciesielski et al., 2014) prior to dataset formation and questionable profiles not caught by the criteria in Table 1 were excluded. The final number of observations for temperature, relative humidity, and winds for the three release sites is listed in Table 2. In the analysis that

follows, data from each of the three release sites was treated independently to highlight the structural similarities and differences between sites.

## 3 Methodology

### 3.1 Introduction to PCA

Principal Component Analysis is a statistical method and dimension reduction technique which transforms correlated variables into orthogonal, linearly uncorrelated principal components (PCs). The PCs are eigenvectors resulting from the eigenvalue decomposition of the correlation or covariance matrix of the original variables. Each individual observation in the time series will have a corresponding PC-weight. These weights take on both signs and represent the specific PC-space coordinate of a discrete data point. PCA is a widely used technique in the atmospheric sciences and a rigorous treatment of

the underlying mathematics and method utilizations can be found in the textbook *Statistical Methods in the Atmospheric Sciences* (Wilks 2011). While typically applied to horizontally distributed (and gridded) data, this study performs PCA calculations on the vertical atmospheric column from the time series of sounding profiles generated in Section 2. Execution of PCA calculations in this fashion has a precedent in the remote sensing community, whereby vertical PCA is used in infrared satellite retrievals as a noise reduction, data compression, and cloud filtering procedure (Huang and Antonelli, 2002;

Smith and Taylor, 2003; Tobin et al., 2007). It has also been used to explore variability in aerosol vertical profiles (Chew et al., 2013; Reid et al., 2016c). Here, column PCA filters noise, but also removes spurious correlations between the artificially




imposed 10 hPa atmospheric layers. The remaining underlying structures exhibit only the dominant signals of variability in the column and the relationships between these variable layers.

### 3.2 PCA Input Data

Temperature, relative humidity, and the U and V wind components were analyzed individually, with each 10 hPa layer treated as an input dimension. In PCA, either the correlation or covariance matrix can be used as input data. The correlation matrix approach assigns a unit variance, and therefore an equal weight to each dimension and is generally used on data with differing units of measure. Conversely, the covariance matrix emphasizes the maximization of variance by each PC. For this analysis the covariance matrix was selected over the correlation matrix, because each thermodynamic variable was run separately and contains only one unit of measure. Furthermore, the purpose of this study is to represent tropical variability, and the covariance matrix method accentuates that variability by attaching a larger weight to more variable dimensions.

Input data may also be standardized before computing the correlation or covariance matrix, and again the decision depends on the emphasis of the analysis. Leaving the data unstandardized inherently tags dimensions with larger ranges as more variable. While this approach is intuitive, relatively smaller fluctuations and correlations between layers will be overshadowed by those with a larger spectrum of values (Fig. 2). For instance, the overall range of tropical temperature values is small from the surface to the tropopause at 150 hPa, beyond which temperatures fluctuate more. This relatively larger temperature range will be the first signal captured by the PCA algorithm (Fig. 7).

Conversely, standardizing the input data weights layers with diverse ranges equally and exposes more fine scale system relationships and vacillations. Additionally, the non-standardized matrix retains skewness found in the original dataset, which is apparent in the distribution of PC weights. As an example, histograms of the RH PC-weight distributions are shown in Figure 3, along with the associated distribution fit, generated via the non-parametric kernel density estimation method. It can be seen that, as with the variable itself, the descriptive statistics for RH are also non-normal. The choice to standardize or not will depend entirely on the intended use of the structure functions and whether the signal itself or its amplitude is more important. In this study a non-standardized methodology was applied to underscore the relevance of magnitude in the representation of variability.

In summation, four 96 x 96 covariance matrices of non-standardized values of T, RH, and U and V winds were generated for each sounding site. The eigenvalue decomposition of the non-standardized covariance matrices was performed to transform and reproduce the radiosonde data in principal component space. The resulting orthogonal eigenvectors (PCs) are axes indicating a mean independent signal and represent a frame around which values (PC-weights) may fluctuate. Due to the nature of eigen-decompostion, the exact sign of the PCs is irrelevant and can be inverted (Figure 6). Furthermore, because scalar multiples of eigenvectors are also solutions to the eigen-decomposition, the eigenvectors were scaled to have a unit length of one. This ensures that all PCs of the same variable have identical lengths for comparison and differ only in direction. In all, the PCs identify atmospheric layers with strong variability, express the extent of this variability, and describe connections between the 10 hPa slabs throughout the atmospheric column.





### 3.3 PC Retention Analysis

Although an inherently subjective decision, the number of PCs, or dimensions, to retain was based primarily on the percentage of total variance explained by each PC (Table 3), found by dividing the eigenvalues by the total variance. However, three additional factor tests were applied (Figure 4) based on Kaiser's Rule (Kaiser, 1960), Cattell's visual scree

plots (Cattell, 1966), and Horn's Parallel Analysis (Horn, 1965). Kaiser's Rule retains any eigenvector with an eigenvalue greater than 1.0, and tends to overestimate the number of significant PCs (Zwick & Velicer, 1986), making it useful only as an upper bound. Horn's Monte Carlo simulation process is often considered the most accurate (Zwick & Velicer, 1986; Thompson & Daniel, 1996), whereby PCs are significant only if the eigenvalue is larger than the 95th percentile of the distribution of eigenvalues derived from random normal uncorrelated data (Cota et al., 1993; Glorfield, 1995). The parallel

analysis test was carried out with a program similar to the one found in Ledesma and Valero-Mora (2007). After this assessment, it was determined that only five PCs needed to be retained to represent a majority of the variance in the tropical MC system.

Despite the reduction in dimensions, the five retained components are capable of reconstructing the tropical atmosphere with an order of magnitude less input data. By applying the vector direct product between the eigenvectors and the PC weights

and adding the layer mean, a specific observation can be reconstructed outside of PC-space and in its natural unit of measure. If all PCs are retained, this transformation will flawlessly reproduce the original data. As the number of PCs is reduced, the conversion will become an increasingly coarse approximation of the original observations. Examples of 5-PC reconstructed soundings are included in Figure 5, along with the original radiosonde data for comparison. The fundamental thermodynamic patterns are easily reproduced using just 5-PCs, including soundings that are close to moist adiabatic (Fig. 5a) and relatively

broad dry layers (Fig. 5b). In soundings with more fine scale structure, the 5-PC approximation cannot reproduce rapid vertical changes in the original data. Strong inversions (Fig. 5c) and fine scale humidity variations (Fig. 5d) may be missed or will be heavily smoothed, and additional PCs would be required to represent these features. Nevertheless, the retained components do capture the bulk characteristics of the soundings, and clearly represent a majority of the variance explained in the tropical thermodynamic system (Table 3). As a result, even in soundings with complex vertical thermodynamic

structures the 5-PC approximation is capable of capturing the overall shape of the sounding and reproducing the salient features of the observed atmospheric environment.

### 3.4 PC Rotation

Because PCs are necessarily orthogonal, each has a strong modal dependency whereby the number of inherent modes corresponds to the PC number (Fig. 6). While the system information is still retained in PC-space, this enforced modality

makes assigning physical interpretations difficult, if not impossible, after the first component. To aid understanding, the axes can be rotated to remove the modality signal and make patterns more pronounced. The original PCs contrast the rotated components (RCs) in their depiction of modality. The PCs have varying modes based on the component number, whereas the





RCs have five modes each, but result in one accentuated mode with RC weights which are either large or near zero. Figure 7 has been included as an example to highlight the difference in modality in the PCs and RCs. Figure 7 also illustrates the fact that positive and negative eigenvectors are possibilities for both the PCs and RCs, and that the area between the two represents the full spread of variability. It is clear from this plot that the non-rotated PCs are constrained to have increasing

5 numbers of modes of variability with increasing PC number. In contrast, the RCs are allowed multiple modes, with a primary mode highlighted in each component. The specific modes of variability, and their physical interpretation, will be discussed momentarily in Section 4.1.

The total variance explained by the RCs will be identical to that of the original PCs, but the variance explained by each RC is spread amongst the retained components and no longer weights primarily on the first component. Because of this

10 transformation, each RC has virtually equal importance, which is opposite of the variance explained in Table 3. The Varimax rotation method was selected for this application, since it retains orthogonality and the uncorrelated nature of the components. Moving forward, all analysis in this study will focus on the RCs for more interpretable results.

### 3.5 Interpretability and Limitations

It is important to note that levels at which the structure functions are non-zero indicate layers of variability in the

15 atmosphere. However, due to the reduced dimensionality of PC-space, certain limitations are imposed on the results. For instance, there is no indication of the physical atmospheric phenomenon producing the observable variability, nor can the period of these oscillations be calculated. To attach meaning to these signals, the structure functions must be compared to existing climatology (Fig. 9), which will be analyzed in Section 4.1.

Furthermore, T, RH, and the U & V wind components are inspected independently in this study. While in reality these

20 variables are coupled, their interconnectedness is outside the scope of this paper. Because of the independent treatment of thermodynamics and wind, the signal magnitude is unique to its respective variable and cannot be compared across variables. This is a consequence of the structure function axes being in a standardized unit. The magnitude on the x-axis reflects the individual variable's descriptive statistics (Fig. 2). As such, while the signal magnitude may be identical in normalized PC-space, the mapping of that signal onto the variable's correct units may vary widely. This is intuitive, because

25 the unit of measure for T, RH, and wind are dissimilar, as are the range of associated values in the tropics.

Nevertheless, looking at results in a lower-dimensional subspace has its advantages. Firstly, the structure functions are orthogonal and independent signals, suggesting unique phenomena are captured by each component. Additionally, within a component the relationships between layers are apparent. This information is invaluable in that it provides a framework for how perturbations at a specific level affect the rest of the atmospheric column, along with the relative sign and magnitude of

30 this effect. Moreover, due to the unit scaling of the eigenvectors, the signal magnitude is comparable across the five RCs for a single variable. While each RC is of equal significance to the total variability of the system, the magnitude is an indication of relative importance.





## 4 Modes of Variability and Analysis

### 4.1 Case Study: Ranai

Although the MC is comprised of many islands, Ranai's location is unique: the minor island is part of the Indonesian Riau Archipelago in the South China Sea and is distant enough from the major MC islands to be largely free of the influence of
land and the diurnal cycle of convection forced by day-time heating of the surface. As such, the Ranai site is essentially maritime and can therefore provide a basic framework for examining tropical over-ocean atmospheric variability. We will use the structure functions obtained from Ranai as a baseline for comparison with Puerto Princesa and Singapore, which are increasingly land-influenced, respectively (Section 4.2, Fig. 10).

Structure functions (RCs) for each sounding variable at Ranai are plotted in Figure 8. The first RC in relative humidity
appears to reflect deep convective moistening/drying of the upper troposphere. This strong signal represents the oscillation between a saturated upper atmosphere during convection and a dry region of subsidence preceding and following the wave disturbance. The structure of RC1 also indicates that upper tropospheric humidity covaries with vapor content at lower levels. Increases (decreases) in RH between 300 – 400 hPa are positively correlated with increases (decreases) in RH in a shallow layer between 600 – 850 hPa, and anti-correlated with decreases (increases) in RH around 500 hPa. The two lower
tropospheric modes are proportionally smaller than the dominant deep convective signal. RC2 in RH reflects the moistening and drying of the lower free troposphere, RC3 represents deep convective detrainment in the upper troposphere/lower stratosphere, RC4 captures melting level water vapor detrainment, and RC5 reflects variability associated with shallow convection atop the marine boundary layer. It is important to note that more variability may exist within humidity in the upper troposphere and lower stratosphere, but its detection is outside the current limitations of radiosonde sensors (Soden
and Lanzante, 1995). RC1 for temperature appears to represent the seasonal shift in the ITCZ, which causes a cyclic change in temperature above 150 hPa. RC2 reflects warming in both the boundary layer and the upper troposphere, perhaps a reflection of the connection between lower tropospheric warming (and moistening) and the incidence of deep convection. RC3 appears to be a deep convective heating / radiative cooling mode, while shallow convective heating is represented in RC4. RC5 appears to reflect the signal of upper-tropospheric heating. Examination of the winds reveals cyclic changes in the
zonal westerlies aloft (RC5; note the lack of a corresponding structure in the RCs for the meridional wind). Fluctuations possibly associated with deep convective detrainment may be seen in RCs 2 and 4, as well as lower tropospheric shifts associated with fluctuations in the monsoon trough (RC3). RC1 is intriguing, and suggests covariability between upper tropospheric (~100 hPa) and mid-tropospheric (~400 – 500 hPa) zonal winds. A similar structure may be seen in RC3 for the meridional winds, and it is possible that this reflects simultaneous detrainment from convection in the upper troposphere and
around the melting level. The overwhelmingly dominant signal in the V-component of wind is the seasonal monsoon. The MC monsoon is characterized by a complete reversal in the north-south wind component at the surface (RC1, RC2) and a corresponding reversal in the upper troposphere of the opposite sign (RC4, RC5).





To aid in attributing meaning to the RCs, pressure versus time contour plots of the radiosonde observations are included (Fig. 9) for the entire time period (2008-2016) and for a subset of that range (2013-2015). Along with the time series, MERRA2 outgoing longwave radiation (OLR) and daily precipitation are included to illustrate the general convective environment. Lastly, the active (Phases 3-5) and suppressed (Phases 6-8) MJO amplitudes for Ranai are plotted. The

distinction between active and suppressed phases was determined based on recent studies by Peatman et al. (2015) and Birch et al. (2016). MJO Phases 1-2 are transition phases for Ranai, do not significantly impact convection, and were therefore not incorporated. Note that the active and suppressed phases are unique to the Ranai release site and are not representative of the entire MC.

Examination of contour time-height plots of RH, T, U, and V help to illustrate the variability seen in the vertical structure

functions. Periodic oscillations in RH in the boundary layer, lower free troposphere, mid-troposphere, and upper troposphere are clearly visible in the top row in Figure 9. Close inspection of the right column of Fig. 9 reveals significant moistening around January 2014, with mid-tropospheric warming prior and cooling following. Cooling in the upper troposphere (anvil-level) with warming below is evident in late fall 2013 as well. The monsoon reversal in the upper-tropospheric meridional wind can clearly be seen in the 2013 – 2015 v-direction winds. Examination of the precipitation, OLR, and MJO index plots

for 2013-2015 indicates low precipitation and high OLR associated with periods during which the MJO was in a suppressed phase and vice versa for the active phases. Finally, an examination of the u-component winds from 2008 – 2016 reflects activity associated with the QBO; downward propagating easterly wind anomalies can be seen starting in mid-2010, early 2013, and early 2015.

**4.2 Release Site Comparison**

As mentioned above, the sounding release site at Ranai is located on a small island generally representative of over-ocean conditions, while the sites at Singapore and Puerto Princesa are surrounded by, or located in close proximity to, larger areas of land (Fig. 1). Singapore, at the Malay Penninsula is adjacent to the island of Sumatra, whose topography and biomass burning events exert an influence on the regional meteorology. Puerto Princesa is located on Palawan Island, which is bordered on the west by the South China Sea and on the east by the Sulu Sea. While Palawan island is long and narrow, it

has enough topography that it may reasonably be expected to exhibit orographic precipitation enhancement. In addition, its location in the northeastern portion of the MC makes it subject to periodic tropical cyclones as well as a portion region of the southwest monsoon. Comparison among the thermodynamic and wind structures obtained from the three different sounding sites allows an assessment of the possible influence of land as well as TCs and different portions of the monsoon trough. A cursory examination of the RCs for each site (Fig. 10) indicates similar patterns of structural variability across the MC, a

reflection of the similarity in the meteorological drivers in the region (primarily convection, convectively coupled waves, and the seasonal monsoon).

Closer inspection reveals important differences among the three sites. First, note that the ordering of RCs is rarely the same among the soundings. Recall that the PCs are arranged in decreasing order according to the fraction of variance explained,



while the RCs have weighting that is approximately similar to one another. Even so, the order of the RCs does reflect decreasing variability, and as such it is useful to compare RC ordering among the sounding sites. To facilitate the comparison, we have chosen to plot similar structures together, regardless of the order of the RC (with Ranai the default). The dominant variability in RH is contained in the mid-tropospheric moist mode, indicative of detrainment from congestus

and deep convection, while the main mode at Ranai is the deep convective mode coupled with shallow moistening from trade cu. Ranai also exhibits more variability in the deep upper tropospheric moistening/drying (RC3), while the shallow moistening is more prominent at Puerto Princesa and Singapore. The rank ordering of RCs of temperature and RH is identical between Singapore and Puerto Princesa with different ordering at Ranai. All three sites exhibit maximum variability in temperature in the upper troposphere, likely reflecting the seasonal oscillation in the height of the tropopause

with the meridional shift in the ITCZ. The secondary mode of variability is also similar among the different sites, with lower tropospheric cooling/warming that positively co-varies with smaller magnitude cooling/warming in the upper troposphere. Overall, Ranai and Puerto Princesa appear to be more similar in structure to each other than to Singapore. The largest differences appear in RCs 3-5; Singapore has larger amplitude variability in the upper troposphere relative to Ranai and Puerto Princesa, and the upper-tropospheric mode (far right panel, Fig. 10) is located higher than at the other two sites.

Interestingly, the u-direction wind RCs (Fig. 10; second row from bottom), while exhibiting similar structures among all three sites, have different rank order among the sites. Finally, the ordering is identical between Ranai and Singapore for the meridional winds (Fig. 10, bottom row), while RCs 1 and 5 are reversed for Puerto Princesa. Recall that Ranai RC1 reflects the low-level oscillation in the meridional wind, while RC5 reflects the oscillation in meridional winds in the upper troposphere. It appears that, at Puerto Princesa, there is more variability in the meridional upper tropospheric flow than there

is at low levels.

While the rank ordering of the RCs varies among the three sites, we wish to point out that the form of the vertical structures obtained from the rotated PCA at three rather widely dispersed sounding sites are very similar. The implication is that (1) similar patterns of variability in both thermodynamic and dynamic vertical structure occur across the MC, and (2) the drivers of this variability (e.g., the spectrum of convective waves and seasonal, intraseasonal, annual, and inter-annual

modes) are expressed similarly across the MC.

### 5 Conclusions and Applications

This study applied Principal Component Analysis to 2008-2016 radiosonde time series data to transform vertical patterns in temperature, relative humidity, and winds into structure function profiles for the Maritime Continent. The main benefits and conclusions from this decomposition of observations into a lower dimensional subspace can be summarized as follows:

1. PCA reduces system noise and condenses a large spectrum of meteorological fluctuations to a few coherent signals. The resulting principal components and rotated components are orthogonal, and represent independent signals of variability in the thermodynamic variables and winds.





2. Structure functions obtained from PCA approximate a majority of tropical thermodynamic variability, and the vertical location of this variability, in a much lower-dimensional sub-space. Specifically, layers with large temporal variability are highlighted, while redundant correlations between levels are ignored, leaving only the fundamental relationships between layers.

3. In most cases, the rotated structure function signals can be attributed to physical atmospheric oscillations and phenomena. Specific signals that are of particular interest include:

- Rotated components in relative humidity show coherent structures indicative of: tropopause level detrainment from deep convection, moistening/drying of the upper free troposphere, melting-level detrainment, and shallow convective moistening/drying.

- Upper tropospheric humidity co-varies with vapor content at lower levels: increases (decreases) in RH between 300 – 400 hPa are positively correlated with increases (decreases) in RH between 600 – 850 hPa, and anti-correlated with decreases (increases) in RH around 500 hPa.

- Temperature structure functions reflect the influence of deep convection and radiative cooling in the upper troposphere, as well as convective heating in the lower free troposphere.

- Vertical structure functions in the u and v components of the wind reflect the seasonal variability in the monsoon, as well as more local effects of convective detrainment at various levels.

4. Structure functions generated from MC sites in very different contexts (remote island, larger mountainous island, and proximal to a larger land mass) are remarkably similar. This indicates that the dominant modes of variability are the result of the propagation of tropical waves and the monsoon, rather than topographic or latitudinal influences,

which may exist in higher order components. The most discernable difference among release sites is the position of the tropopause, and the fact that different structures explain a different fraction of the overall variance among the three different sites.

Overall, the application of PCA to the vertical atmospheric column and subsequent transformation of vertical thermodynamic and wind information into principal components expresses the important variability and relationships among

layers. While this study focused on the Maritime Continent, the same methodology can be applied to different variables, time durations, and radiosonde release sites. In addition, these structures can be used to examine co-variability between thermodynamic and dynamic structures and measures of convective activity (e.g., precipitation rate and OLR). They can also be used to control for variations in meteorology when examining temporal and spatial variability in aerosol content and composition.

30
Undoubtedly, the future applications of this method are far reaching. With short-term tropical meteorological variability constrained, it is possible to identify relatively elusive system signals in this region. Furthermore, because of their unique properties, the structure functions can be used as boundary conditions in numerical models. Especially important for modeling is the representation of the interconnectedness between levels and the scaling of perturbations throughout the



atmospheric column. In all, decomposing vertical patterns into modes of variability will allow for more realistic future model simulations and analysis of atmospheric processes over the MC.

### Acknowledgements

Funding for this project was provided by the NASA-Inter-Disciplinary Science grant NNX14AG68G. Ms. Bukowski's and
Mr. Atwoods' participation was also supported by the Naval Research Enterprise Internship Program (NREIP). Special thanks to Dr. Paul Ciesielski at Colorado State University and Dr. Pat Pauley at NRL-Monterey for their radiosonde expertise, along with the University of Wyoming for access to and decoding of upper-air observations. We are also grateful to Jannet Noriega (NRL Science and Engineering Apprenticeships Program) for her help quality assuring the radiosonde data used in this study. Final thanks to NASA for access to the MERRA2 dataset and the Climate Prediction Center for MJO data
access.

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





| Sounding Quality Control Flags | | |
|---|---|---|
| **Variables** | **Level** | **Threshold** |
| Temperature | 1000-hPa | 15° C < T < 40° C |
| Relative Humidity | 1000-hPa | 30 % < RH < 100 % |
| Temperature | 480-hPa | T < 0° C |
| Temperature | 300-hPa | T < -20° C |
| Temperature | 200-hPa | T < -40° C |
| Temperature & Dew Point | All | T > Td |

**Table 1: Quality control flags enacted during the sounding dataset creation as a first checkpoint in removing erroneous soundings or raw data decoding. This list is not comprehensive, as a hand analysis was performed after these flags were applied.**





| Number of Soundings (2008-2016) | | | |
|---|---|---|---|
| Variable | Ranai | Puerto Princesa | Singapore |
| Temperature | 3429 | 2326 | 4143 |
| Relative Humidity | 1401 | 2235 | 2878 |
| U and V Wind | 3380 | 2257 | 4096 |

Table 2 Total number of quality controlled radiosonde observations for each of the release sites. To increase the number of data points, each variable was given independent consideration during the quality control process.





| Percent Variance Explained by Principal Components | | | | | | | | | | | |
|---|---|---|---|---|---|---|---|---|---|---|---|
| | Ranai | | | | Singapore | | | | Puerto Princesa | | | |
| PC # | T | RH | U | V | T | RH | U | V | T | RH | U | V |
| 1 | 35.44 | 53.86 | 42.58 | 43.80 | 36.02 | 38.14 | 36.53 | 47.81 | 33.60 | 64.38 | 51.17 | 45.15 |
| 2 | 21.16 | 13.29 | 19.74 | 14.52 | 22.23 | 17.35 | 19.59 | 12.03 | 20.50 | 10.27 | 25.68 | 21.43 |
| 3 | 10.00 | 10.34 | 9.58 | 10.29 | 10.49 | 10.16 | 10.38 | 10.32 | 8.51 | 5.82 | 5.41 | 8.50 |
| 4 | 6.73 | 4.83 | 7.72 | 8.61 | 6.20 | 6.67 | 9.28 | 7.62 | 8.16 | 3.68 | 4.98 | 6.79 |
| 5 | 5.52 | 3.96 | 6.41 | 5.43 | 5.19 | 5.40 | 7.77 | 5.61 | 6.29 | 3.06 | 3.49 | 4.38 |
| Sum | 78.85 | 86.28 | 86.02 | 82.65 | 80.13 | 77.72 | 83.55 | 83.39 | 77.06 | 87.20 | 90.73 | 86.25 |

**Table 3 Percent of total variance explained for each variable's PCs from the three release sites. The first PC represents the highest variance and each subsequent component explains a decreasing amount of the residual variance.**



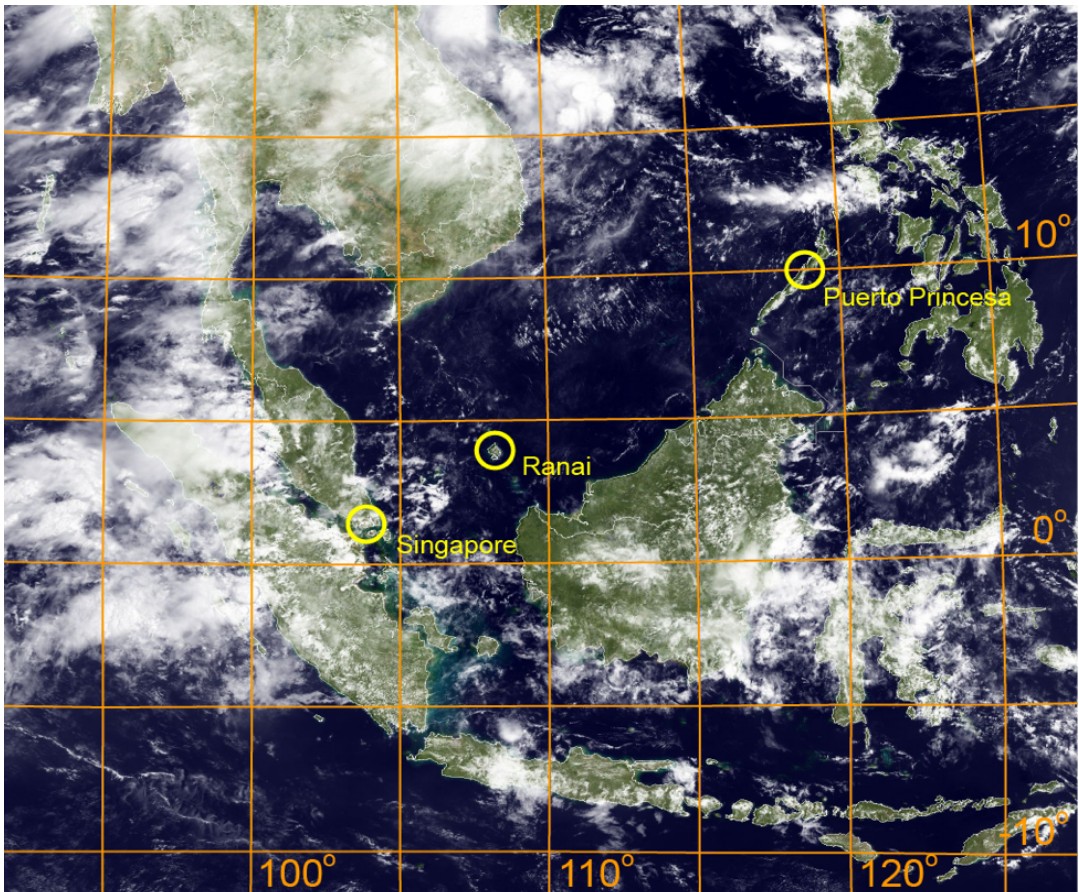

**Figure 1: MTSAT visible image over the Maritime Continent obtained at 0332 UTC 14 September 2011. Sounding sites used in the analysis are labeled and circled.**





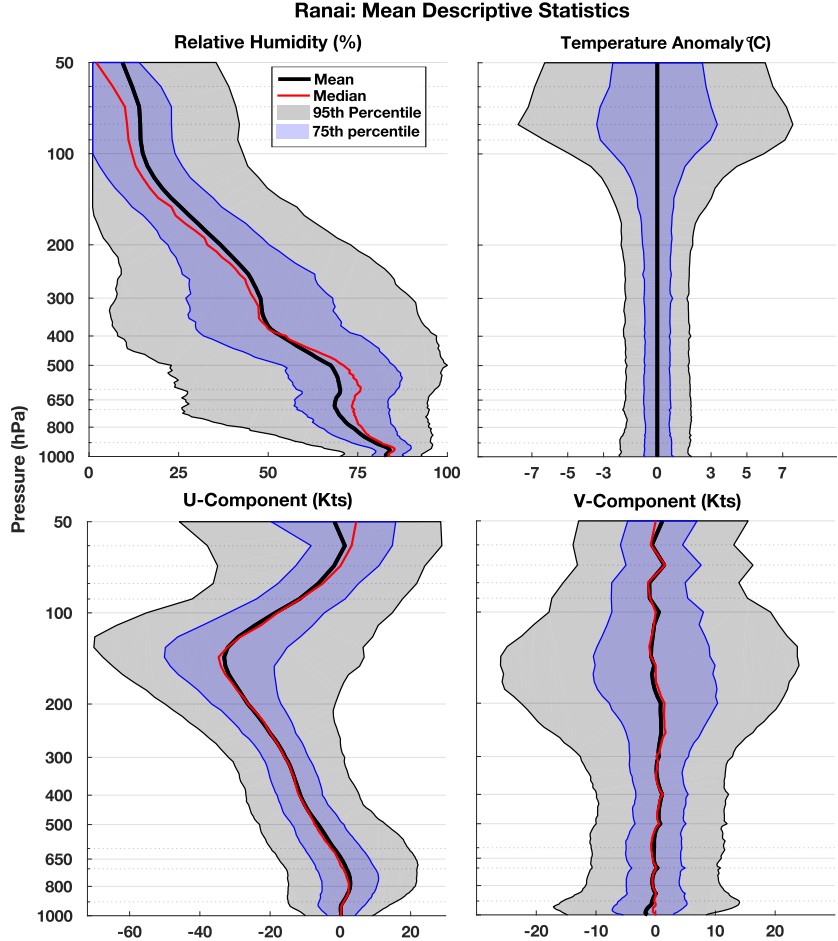

**Figure 2: Mean, median, interquartile range, and 95th percentile range of the Ranai sounding dataset. Relative humidity and the U-component show signs of skewness. The range of values for each variable's unit of measure at each level is apparent.**





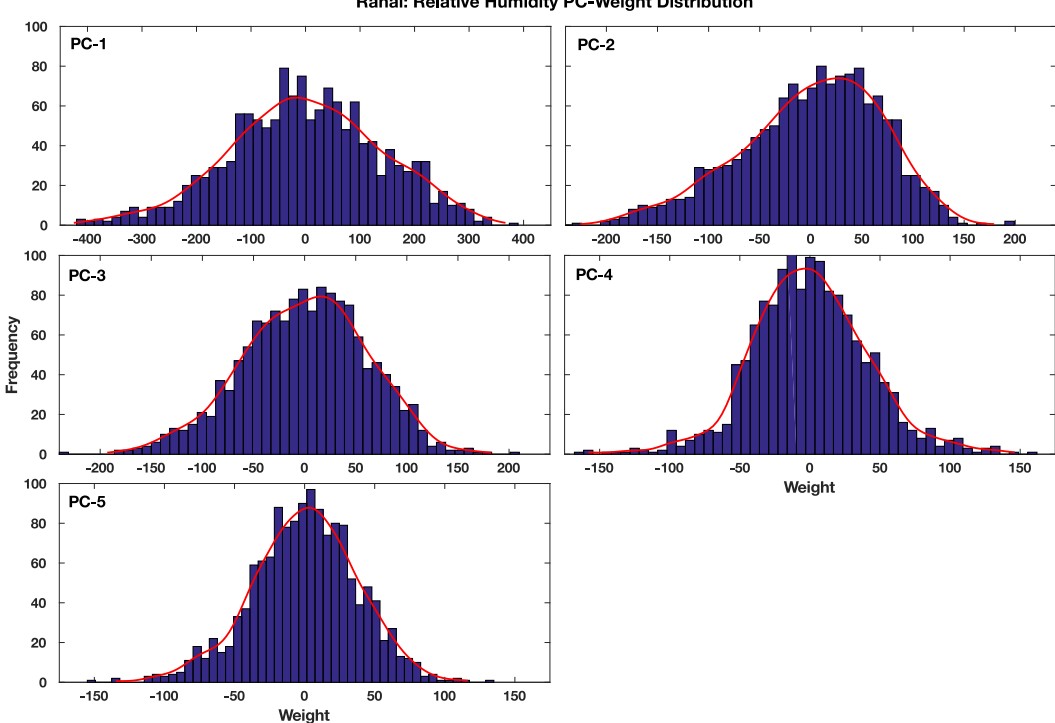

**Figure 3: Histograms of the relative humidity PC-weights for Ranai in blue, with distribution fits in red. The original descriptive statistics (Fig. 2) showed signs of skewness for this variable. Because the original input data was not standardized, the PCA algorithm retains the inherent skewness, which is observable in PC-2, PC-3, and PC-4.**





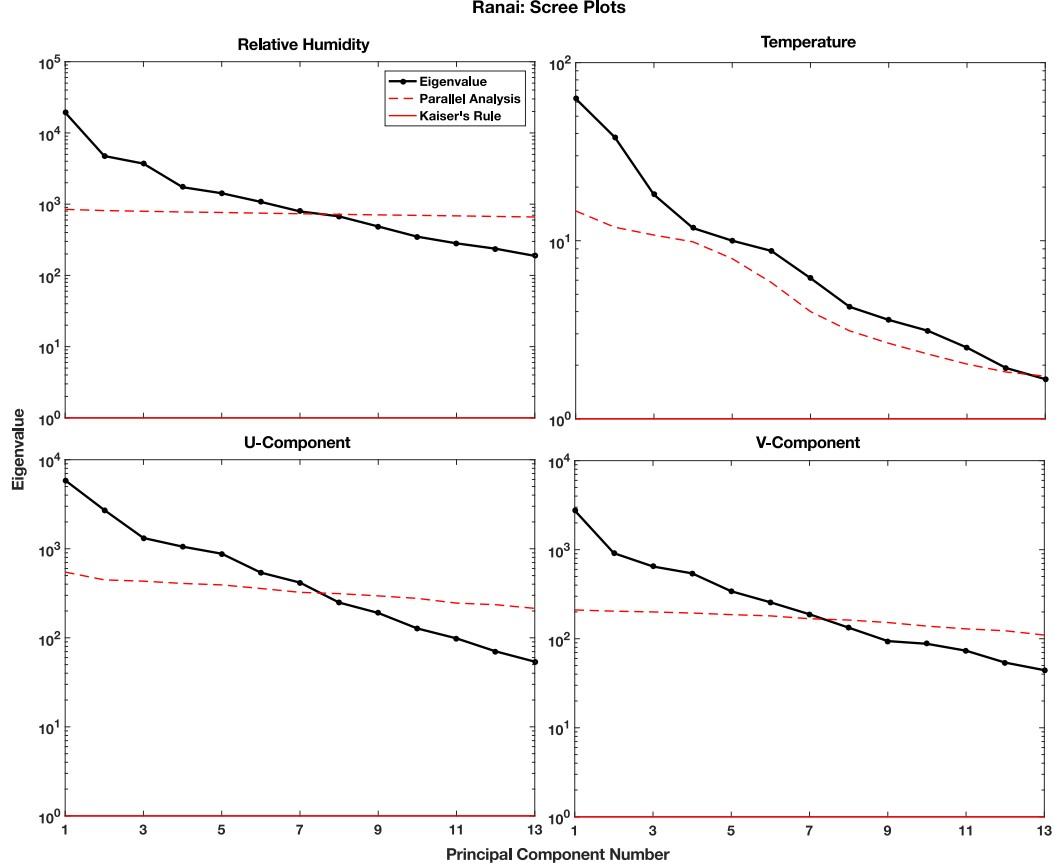

**Figure 4: Scree plots for the Ranai release site for the four variables of interest. The calculated eigenvalues are in black and the PC retention criteria are in red. Generally, Kaiser's Rule (solid red) overestimates the number of PCs to retain, whereas Horn's Parallel Analysis (dashed red) is more reliable.**




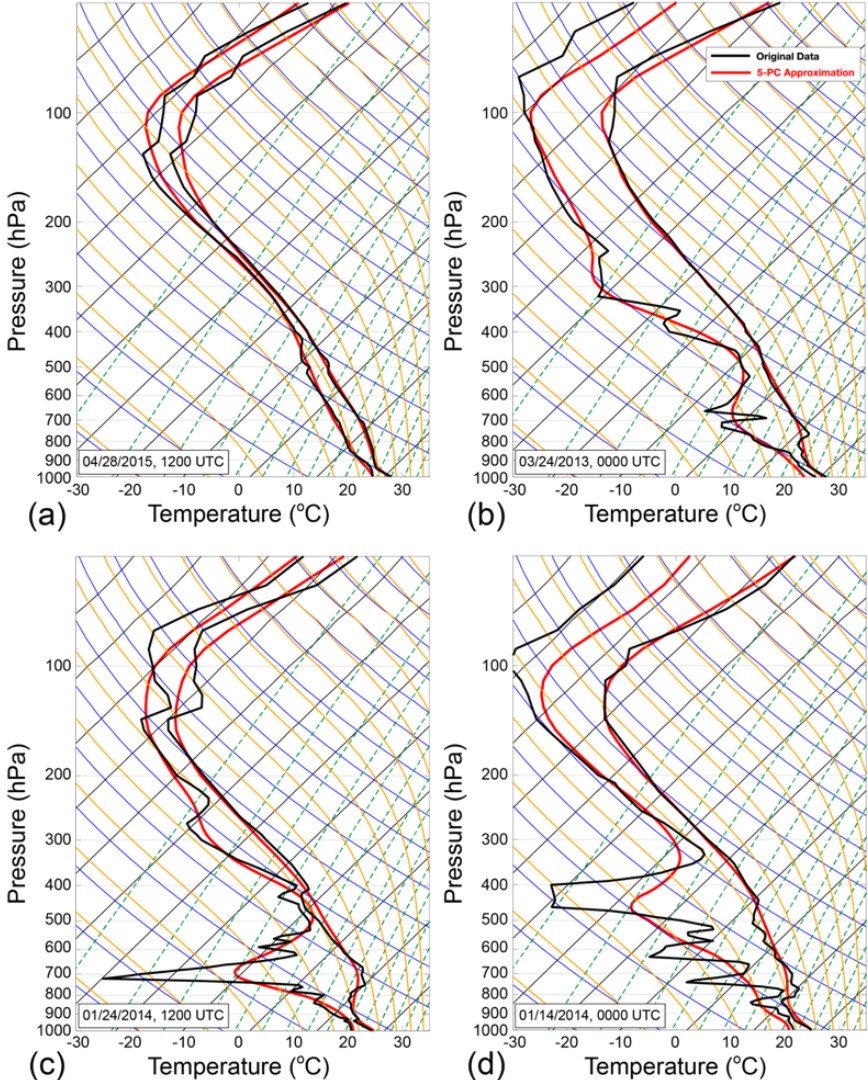

**Figure 5:** Original radiosonde observations (black) from the Ranai release site and reconstructions of the same radiosonde data using only 5 principal components (red). Top rows (a and b) represent thermodynamic structures for which the 5-PC approximation performs well. The bottom row (c and d) depicts soundings with more complex vertical structure that are more difficult to reproduce with a small number of retained components.



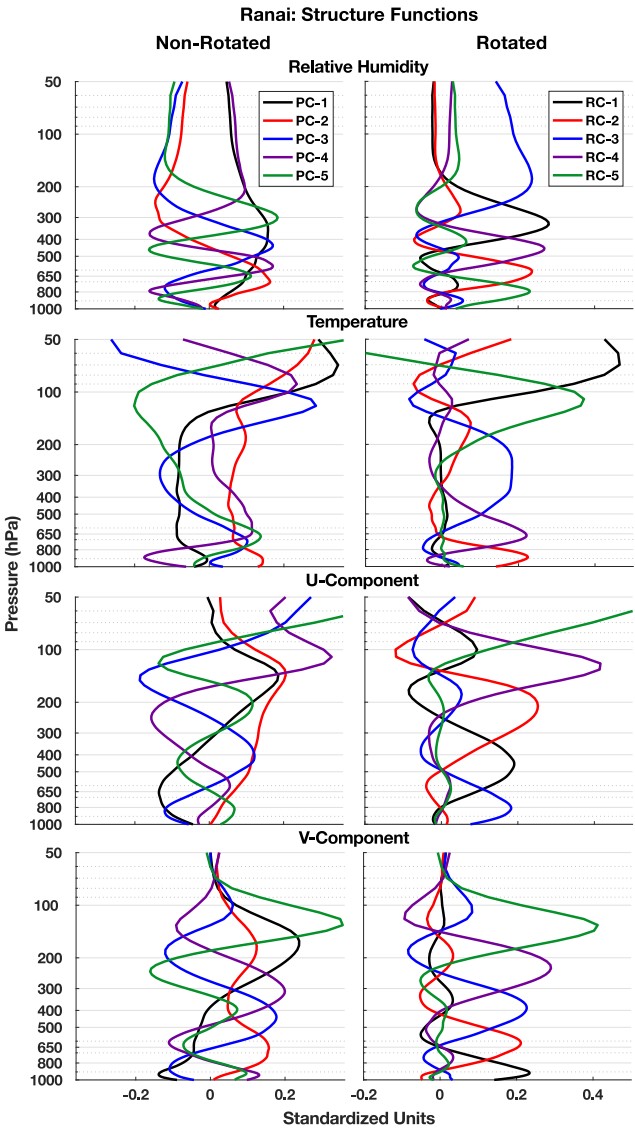

**Figure 6:** Non-rotated PCs (left) and rotated components (right) for all variables at the Ranai release site. Only one sign of the eigenvector is plotted. The strong modality (sign change) is apparent in the PCs, whereas the RCs have five modes total due to the Varimax rotation, but only one mode contains a strong signal.





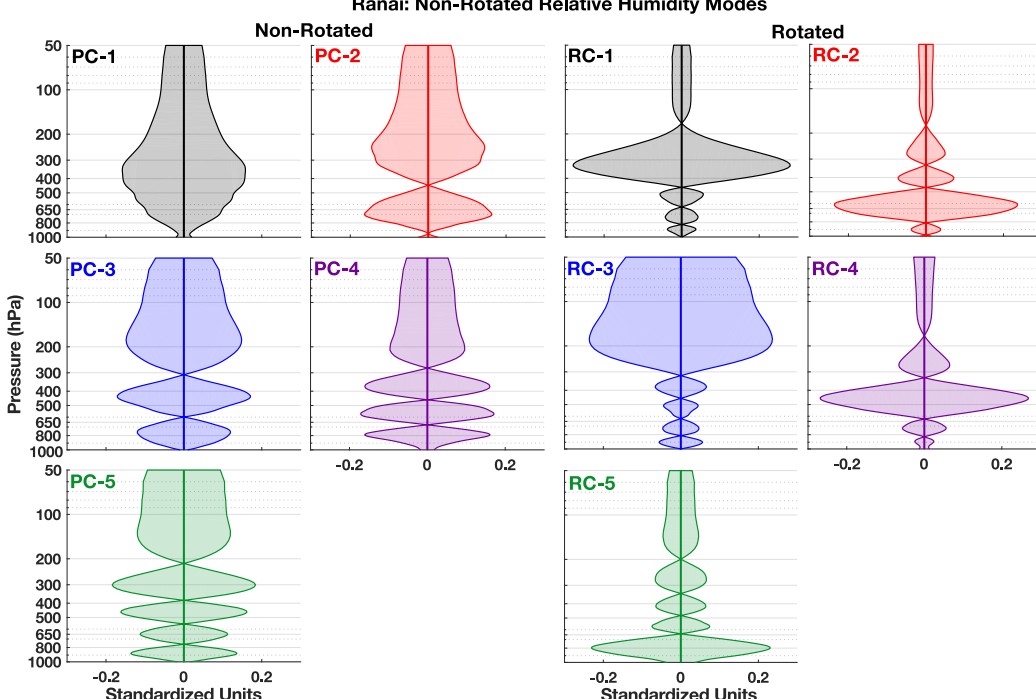

**Figure 7: Non-rotated PCs (left) and rotated relative humidity RCs (right) at the Ranai release site. Bold lines indicate the zero line and the positive and negative values of the eigenvectors. The shaded area represents the total spread of variability.**





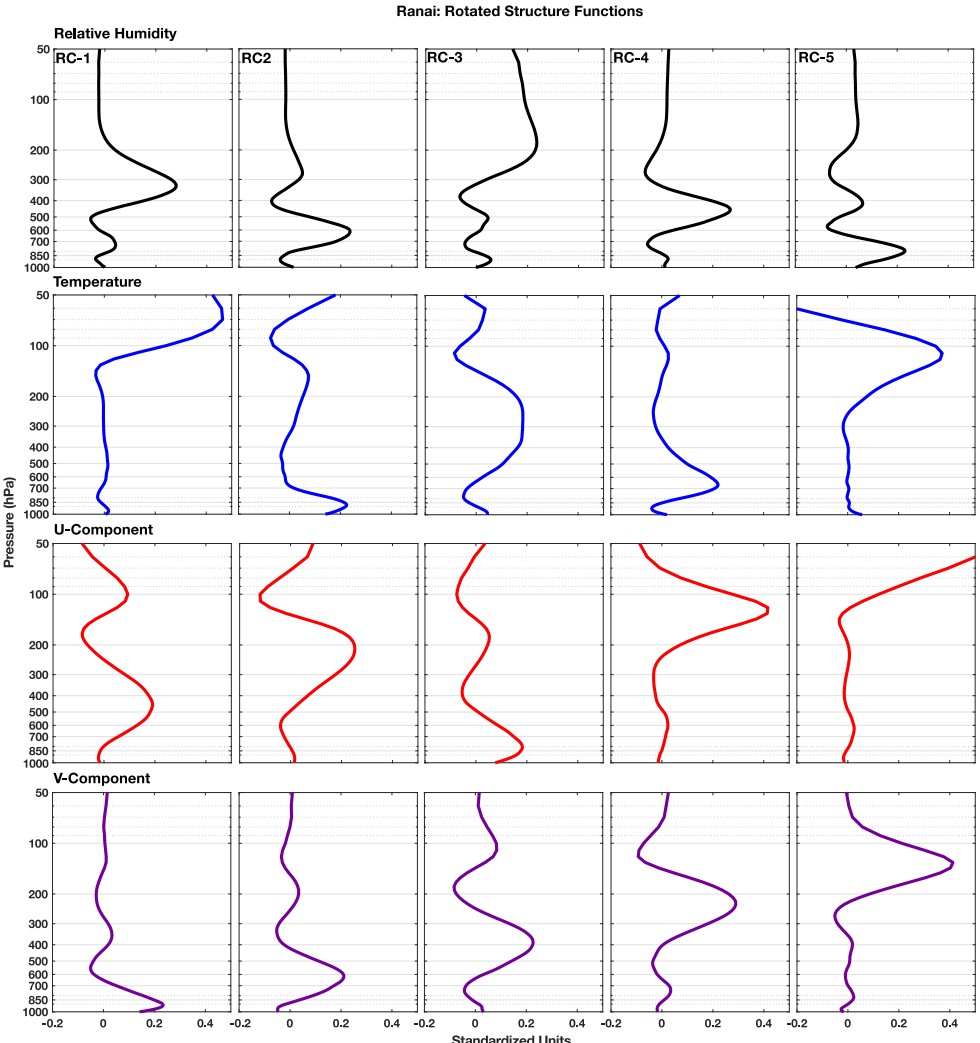

**Figure 8: The first five RCs at the Ranai release site for relative humidity (black), temperature (blue), U-component (red), and V-component (purple). The y-axis is in log-pressure (hPa) and the x-axis is in standardized units. For simplification, only one sign of the eigenvectors is included in this figure.**



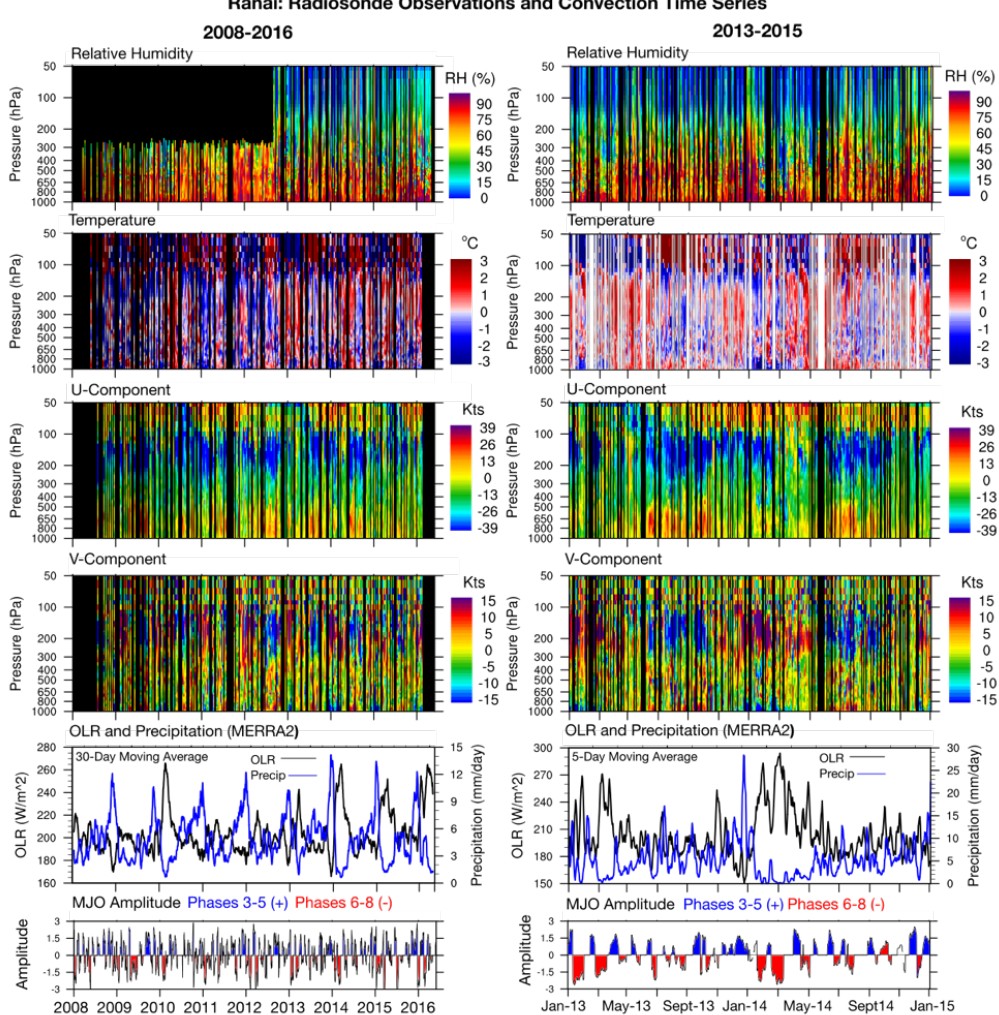

**Figure 9. Time series of temperature, relative humidity, U and V winds, outgoing longwave radiation, precipitation, and the MJO active and suppressed phases at Ranai. The left column is the entire study period 2008-2016, and the right column is a subset of that time period from 2013-2015. T, RH, and wind data in the first four rows is from the radiosonde observation dataset in Section 2. The Ranai radiosonde equipment reported RH only below the 300 hPa level until 2012 and is therefore missing. Areas of the time series shaded in black in the first four rows represent missing observations or dates that did not pass quality control. OLR (black) and daily precipitation (blue) in the 5th and 6th rows come from the MERRA2 dataset. To reduce noise, OLR and precipitation have moving average filters applied. The MJO amplitude was split into active phases (red), which enhance precipitation in Ranai (Phases 3-5), and suppressed phases (blue), which suppress precipitation in Ranai (Phases 6-8).**





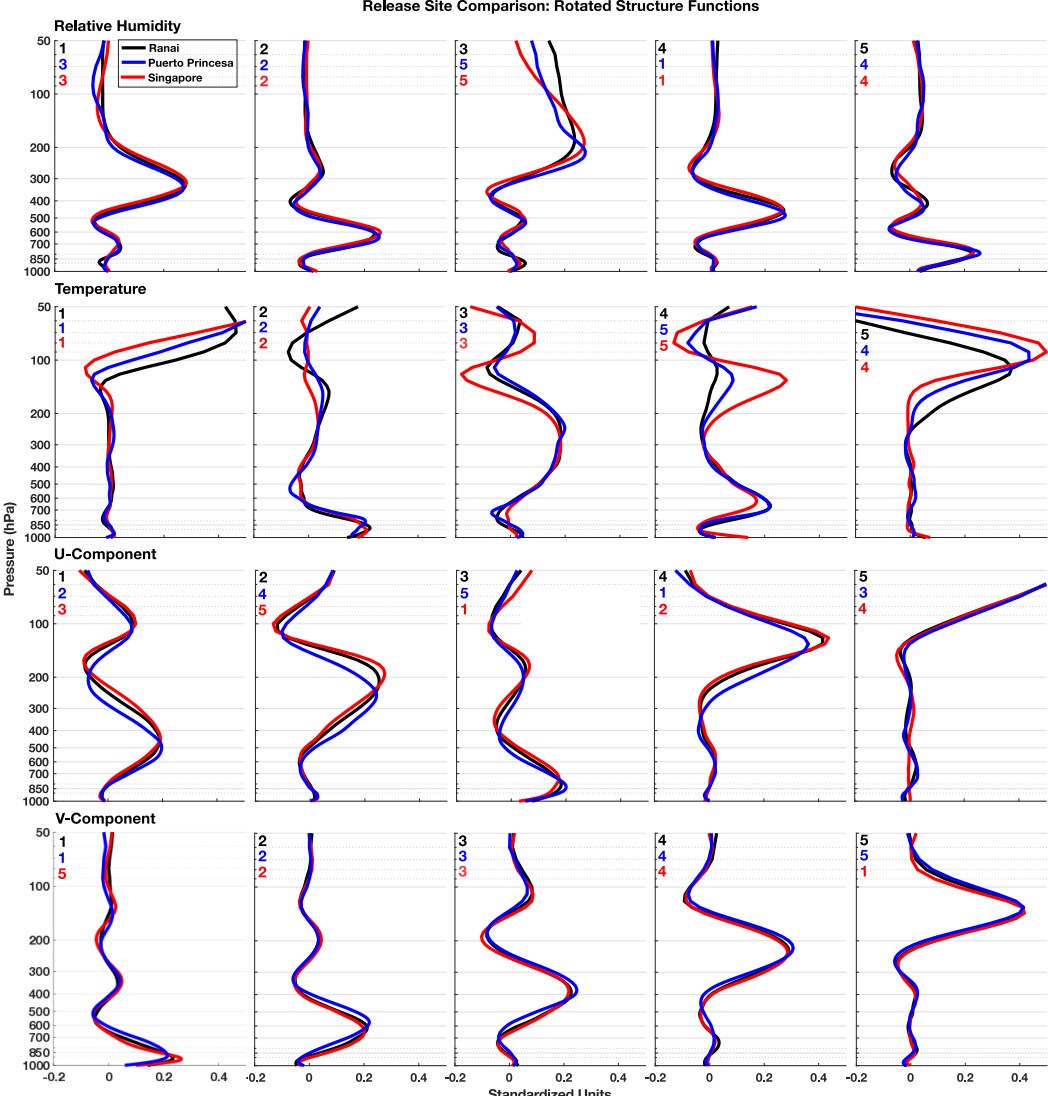

**Figure 10.** RCs for the three radiosonde release sites in Ranai (black), Puerto Princessa (blue), and Singapore (red). The panel numbers represent the RC number for that specific signal and are color coded to match their respective release site.