# Peer review of "Modes of Vertical Thermodynamic and Wind Variability over the Maritime Continent"

_Atmospheric Chemistry and Physics, 2016_

## Referee Comment (RC1) · Anonymous Referee #1 · 28 Nov 2016

General comments: The papers examines temporal and spatial modes of atmospheric variability over the Maritime Continent (MC) by applying principal component analysis to upper-air data from three "representative" sounding sites. With this approach they conclude that the fundamental modes of spatial and temporal variability of this region can be captured with small set of coherent structures. The manuscript is reasonable well written and organized but could you use some additional clarification in several places as noted in comments below. In addition, I was left wondering if the three sites examined, all of which border or are within the South China Sea, are truly representative of the entire the MC region which spans over 60 degrees of longitude and contains over 17,000 islands. It's unclear to what degree the RC analyses of these three sites capture the major modes of variability in this large MC region? What measures did the authors take to conclusively demonstrate this point? Were other sites examined, say

sites bordering the Java Sea or coastal sites in Borneo or Papau New Guinea where topographic effects and the diurnal cycle dominate substantially more than the three sites examined here (see Peatman et al. 2014, QJRMS) or ENSO effects might be more prominent? Short of such efforts, the authors may need to temper their conclusions somewhat to reflect the more limited scope of their analyses. While it might be beyond the scope of this study, it would be helpful to put the variability of these MC sites in context by showing how they differ from sites in the Indian Ocean and West Pacific (e.g, Gan and Manus) where MJOs often typically initiate and dissipate and even a midlatitude continental site which should show dramatically different structures. Another natural extension of this work would be a PC analysis of rainfall at each site to better understand the relationship of the RCs presented in this paper to convection.

Specific comments:

Page 3, lines 23-24: Upon examination of Ciesielski et al. (2014), which documents the sounding types used at the sites in question during the DYNAMO experiment (October 2011 to March 2012), they note that the Singapore site changed sonde types during DYNAMO. Did the other sites use the same sonde type for the entire period examined (2008-2016)?

Page 3, line 24: In what fields are the constant bias corrections available (e.g., humidity, temperature)? Do biases remain constant when the sonde type changes such as occurred at Singapore in late December 2011. Is there any evidence that the characteristics of the RCs changed at Singapore when they switched sonde types?

Page 5, line 14: I'm assuming you mean "temporal range"? Please clarify.

Page 8, line 6 and page 9, line 20: Past studies (e.g., Houze et al. 1981 MWR, Ciesielski et al. 2006 MWR) would suggest that during certain seasons, the Ranai site is not entirely free of the diurnal cycle or the influence of land convection.

Page 8, line 24: are the wind reversals noted here a reflection of the QBO?

Page 9, line 7-8: Please clarify what you mean that the active and suppressed phases are unique to Ranai and not representative of the entire MC.

Page 9, line 9-10: In addition to presenting the time-height plots, which are informative, showing the temporal behavior of the RCs might help clarify what physical phenomena the different RCs are associated with. For example, does RC1 for temperature vary seasonally (page 8, line 20), is RC5 for zonal wind associated (page 8, line 25) with the QBO time-scale changes, do variations in RC1 and RC2 for meridional wind vary on a monsoonal time scale as suggested, and so on?

Page 9, lines 11-12: Are the moistening and the mid-tropospheric temperature changes noted here associated with an MJO?

Page 9, line 25: How high is the topography on this island? How does this compare with other sites in Indonesia that might be impacted by steep topography? This relates back to my earlier comments on the representativeness of this site.

Page 9, line 26: What is meant by a "portion region of the southwest monsoon"?

Fig. 9. Is perturbation temperature being plotted here? If so, please specify in figure caption. What are the differences between the black and white spaces? Would it be possible to linearly interpolate in time the missing data, at least for reasonably short time gaps, shown in the right-hand panels? This would make the structures referred to in the text refers to easier to identify. In the last sentence of the caption are colors red and blue reversed.

Is any ENSO time-scale signal detected in the RCs?

Page 10, lines 9-10: Does RC1 of temperature shown in Fig. 10 reflect differences in mean tropopause height of these sites with Singapore have the highest mean height and Ranai the lowest? If so, would it be possible to list the mean tropopause heights of these three sites?

In Fig. 1 instead of using some random MTSAT image it would be more instructive

if the long-term mean rainfall map for this region were shown. For example, this map could be based on MERRA analyses or preferably TRMM or GPCP observations which are easily accessible. In this manner the reader could see how the sites chosen were located with respect to regions of major convection.

Page 3, line 27: Please state when RH is first mention in the paper whether it was computed with respect to ice or water at temperatures less than 0°C.

Technical corrections:

Page 1, line 9: continent misspelled.

Page 1, line 13: I suggest changing "sounding release sites" to "upper-air sounding sites" here and other places where this phrase is used.

Page 5, line 29: decomposition misspelled.

Page 4, line 9: Suggest rewording, "... the database was subjectively analyzed in the following fashion."

In Table 1 title and caption change "Control Flags" to "Checks". Also suggest changing "checkpoint in" to "pass at". Finally suggest changing last sentence in caption to: "This is not a comprehensive list as additional subjective QC was applied."

In a few of the figures (Figs. 6, 9 and 10) where lines overlap, it's difficult to distinguish between black and dark blue lines. You might try using cyan instead of dark blue to make the lines more distinguishable.

---

## Referee Comment (RC2) · Anonymous Referee #2 · 6 Dec 2016

Overall this is a well written, intelligent paper that appears technically competent (to someone not well versed in PCA). And I think that through the RC's, the author's have identified something intrinsic in the temporal variability of T, RH, U, and V profiles.

However, I am not really sure what new it brings to the table in addition to the usual procedure of generating T/RH composites about high rain events, and I think the paper oversells how useful this technique is likely to be in the future. For example, from work by Kiladis and others, we already know a lot about the temperature and wind anomalies associated with different types of convectively coupled waves in the tropics. This type of procedure, in which we project anomalies on to the types of physically known propagating 3D convectively coupled waves seems more insightful than the 1D PCA done here, in which there is no attempt to physically separate any of the myriad of influences on a particular profile. Below are some specific comments.

[Figure]

(1) Section 4.1. The justification for the physical interpretations to which the various RC's are assigned is often unclear. For example, "shallow convective heating is represented in RC4". But do we really know what kinds of temperature anomalies are likely to be associated with shallow convective heating? For example, if shallow convective clouds occur more frequently (e.g. are triggered) by the moistening and cooling associated with low level upward motion (likely, especially in the vicinity of deep convection), then perhaps shallow clouds are correlated with low level cold anomalies, and any positive correlation between shallow convection with positive RH should not be interpreted as a consequence of detrainment moistening, but some external dynamically imposed influence. For example, even the net effect of precipitating shallow convection on the RH of a particular level is unclear. It is a residual of the drying associated with induced descent, moistening from detrainment and evaporative moistening, and then a slower dynamical response driven by the geopotential anomalies associated with the convective heating. More generally, causality between T, RH, u, v anomalies in the background atmosphere and convective clouds always goes both ways. There can't be a simple one to one relationships between certain types of T/RH anomalies and certain cloud types or heating profiles, as implied here. (Otherwise it seems to me that convectively coupled waves in the tropics could not exist.)

(2) Similarly, sometimes the RC's for U and V are assigned physical interpretations and again the justification is unclear. E.g. "The overwhelmingly dominant signal in the V-component of wind is the seasonal monsoon. The MC monsoon is characterized by a complete reversal ...". I guess it is not clear to me here what exactly is meant by "monsoon" in a region of such complicated topography, or why it must have these impacts on U and V. For example, the three radiosonde locations are at quite different locations in the Marine Continent, so the dynamical signature of the monsoon must vary between locations, but the RC's of the three locations are the same almost (except for ordering). This is noted in point 4 of the conclusion. Should the "monsoon" have the same dynamical signature in all three locations? Perhaps give some explanation of what is really meant by "monsoon". It seems that the authors have simply defined

a particular RC as a monsoon signature, and then remarked that this RC is the same at all three locations, and then say the monsoonal signature is the same at all three stations. Everything proceeds from the initial categorization. But is this really more than a semantic game? Do you really know for certain what types of large scale dynamical motions are associated with a particular RC? How would you prove this? I realize there is some discussion of this in lines 13-14 of Section 4.1, but this wasn't fully convincing to me.

(3) Figure 9. I found this hard to interpret. Especially there was so much variability in the top 4 panels, that the features discussed in the text were not clear to me.

Overall, the authors have done some interesting calculations. It just isn't clear to me what new physical insights are generated, or how these might be used as diagnostics tests of climate models. It would be useful if the authors made more definitive attempts to establish the basis of their physical interpretations, or if not possible, avoid what I see as over-interpretation.

---

## Author Response (AR1)

To begin, the authors would like to thank the reviewer for their time, attention to detail, and thoughtful insights on the paper and research. Each comment will be addressed point by point.

**Specific comments:**

Page 3, lines 23-24: Upon examination of Ciesielski et al. (2014), which documents the sounding types used at the sites in question during the DYNAMO experiment (October 2011 to March 2012), they note that the Singapore site changed sonde types during DYNAMO. Did the other sites use the same sonde type for the entire period examined (2008-2016)?

*According to the World Meteorological Organization Catalogue of Radiosondes and Upper-air Wind Systems, there is no record of the Ranai (Modem M2K2DC sonde) or Puerto Princesa (Sippican Mark II sonde) sites switching sonde types during the study period (2008-2016).*

15   Page 3, line 24: In what fields are the constant bias corrections available (e.g., humidity, temperature)?

*The reporting biases for temperature and wind are small across sonde type and are generally accepted as is, barring external influences at the site. Bias corrections therefore exist mainly for humidity, which is the most uncertain measurement in radiosonde observations. For the VRS92 sondes (Singapore post 21-Dec-2011), the NCAR radiation bias correction (Wang et*
20   *al. 2013) can be applied to correct for a solar radiation dry bias found in daytime soundings and is a form of mean bias correction. The Modem sonde used at the Ranai site can be corrected with the constant bias technique outlined in Nuret et al. 2008. The Sippican sonde used at the Puerto Princesa site was found to have relatively small RH errors (Wang and Zhang, 2008), along with the Graw sonde (Singapore pre-21-Dec-2011). The biases in the Sippican and Graw sondes were deemed small enough that correction was unnecessary. (Ciesielski et al., 2014). To err on the side of caution, during quality control the*
25   *inaccurate low-level soundings of the Graw sonde, specifically at the Singapore site (Ciesielski et al. 2014), were removed rather than corrected. For the three sites in this study, the sonde type was either reported as precise enough for scientific research, or a form of mean / constant bias correction was available to improve its accuracy.*
*        However, rather than implement the corrections, a mean bias test was performed (Supplementary Figure 1), whereby a positive and negative humidity mean bias were introduced to the entire Ranai dataset from xx hPa to xx hPa. The*
30   *choice of both a positive and negative bias was deliberate to ensure that an inflection point was represented. The structure functions from the original dataset were compared to the perturbed-mean dataset and were found to be identical. Because the current correction techniques seek to modify the measurement mean, the amendments themselves have no impact on system variability. Principal component analysis focuses on quantifying system variability and is indifferent to mean quantities or biases therein. Therefore, the application of humidity bias corrections was unnecessary for this study but can be utilized in*
35   *instances where precise mean values are required.*

Do biases remain constant when the sonde type changes such as occurred at Singapore in late December 2011. Is there any evidence that the characteristics of the RCs changed at Singapore when they switched sonde types?

40   *The Singapore site was also tested to determine if the change in sonde type affected system variability. The Singapore dataset was split in two at the date of sonde replacement (21-Dec-2011) and the principal component analysis was performed three times: once on the full dataset (2008-2016), once on the Graw sonde period (2008-2011), and once on the VRS92 sonde period (2011-2016). The resulting structure functions are almost identical (Supplementary Figure 2), with only a few variations at levels above 200-hPa and near 700-hPa. While slight variations between the components exist, it is not possible*
45   *to attribute this simply to the change in sonde type; it could also be an artifact of natural system variability between the two time periods, rather than instrument discrepancies. To further complicate matters, it could very well be a combination of both the natural climate variability and the change in instrument.*

*To test this further, the Koror release site was split in two at the same date (21-Dec-2011) and the analysis was repeated.*
50   *Koror was chosen instead of Ranai or Puerto Princesa because it has the most reliable release record before 21-Dec-2011 and after. Both the Ranai and Puerto Princesa site are skewed in that they have more observations after 21-Dec-2011 than before, which could distort the analysis. Looking at the right panel of Supplementary Figure 2, the Koror site also has upper level differences in the RH RCs, but is missing the mid-level differences at 700-hPa. Because the Koror site did not switch sonde type, this is most likely natural variability. Owing to the fact that the Koror site also has differences when the data set is*
55   *split in two, researchers are hesitant to assign the RC fluctuations only to the change in sonde type at the Singapore release site.*

Page 5, line 14: I'm assuming you mean "temporal range"? Please clarify.

*Unfortunately, PCA does not possess the capability of quantifying the time scales associated with variability. Time scales must be attributed by researchers once the physical interpretation is complete. In this study we can only seek to quantify magnitudes of variability, and here the use of the word "range" comes from the range of magnitudes (Figure 2). To clarify, temperature values near the surface deviate from the mean only by only +/- 2°C, whereas at 100-hPa the range of values is much larger at approximately +/- 7°C from the mean. By comparing the different limits, it is obvious that temperatures in the upper troposphere and lower stratosphere are more variable than the well-mixed layers of the troposphere. The wording in the paper has been updated to clarify the use of the word "range" to indicate that it refers to magnitudes and not time.*

Page 8, line 6 and page 9, line 20: Past studies (e.g., Houze et al. 1981 MWR, Ciesielski et al. 2006 MWR) would suggest that during certain seasons, the Ranai site is not entirely free of the diurnal cycle or the influence of land convection.

*A statement was added to address that the Ranai site can experience a diurnal convective cycle in December due to the northerly monsoon winds and the sea / land breeze circulations off of Borneo (Houze et al., 1981) and that even in non-monsoon months there is some evidence of a diurnal cycle (Ciesielski and Johnson, 2006). Incontestably, no site, maritime, island, or continental, would be completely free of diurnal effects. However, compared to other land masses in the MC, Ranai experiences relatively fewer. For example, in Figure 18 in Ciesielski and Johnson (2006) the summertime months only show a diurnal difference in precipitation on the order of 2 mm/day at Ranai (108°E), compared to Borneo (110°E – 117.5°E) where the difference can be over 18 mm/day. Despite these impacts, compared to the Singapore and Puerto Princesa sites, Ranai will be relatively more maritime.*

Page 8, line 24: are the wind reversals noted here a reflection of the QBO?

*The variability above 100-hPa in the meridional wind is most likely the QBO (RC5). The pressure level where this variability is detected correlates with the time series in Figure 9 where the QBO wind reversal is evident. Also, see Supplementary Figure 3, where the QBO cycle in the meridional wind is clear in RC5.*

Page 9, line 7-8: Please clarify what you mean that the active and suppressed phases are unique to Ranai and not representative of the entire MC.

*Because the MJO propagates eastward, its effects have a spatial and temporal nature. The categorization of the MJO into 8 phases stems from this cyclic propagation. Depending on location, the convectively enhanced phases of the MJO, or active phases, and the convectively suppressed phases will be different. For example, in Peatman et al. (2015) Figure 3, Ranai has positive daily mean precipitation anomalies in phase 3 and is convectively enhanced, while in the same phase Paupa New Guinea presents negative anomalies and is convectively suppressed. Essentially, specific MJO disturbances will be observed at different times at different sites and this must be accounted for when assigning phase numbers as either active or suppressed.*

Page 9, line 9-10: In addition to presenting the time-height plots, which are informative, showing the temporal behavior of the RCs might help clarify what physical phenomena the different RCs are associated with. For example, does RC1 for temperature vary seasonally (page 8, line 20), is RC5 for zonal wind associated (page 8, line 25) with the QBO time-scale changes, do variations in RC1 and RC2 for meridional wind vary on a monsoonal time scale as suggested, and so on?

*Generally, the temporal signature of variability is lost during PCA. However, some structure of this remains in the RC-weight time series (Supplementary Figure 3). Looking at the figure, for almost all of the RCs, the seasonal monsoon is the dominant cyclical signal. These plots can help attribute meaning to some of the RCs, but we cannot overlook the fact that the monsoon cycle will modify the thermodynamic signature.*

Page 9, lines 11-12: Are the moistening and the mid-tropospheric temperature changes noted here associated with an MJO?

*In Figure 9 it looks like the moistening and temperature changes are mostly associated with the monsoon cycle. However, a MJO event is embedded within the monsoon and would act to further modify the thermodynamics.*

Page 9, line 25: How high is the topography on this island? How does this compare with other sites in Indonesia that might be impacted by steep topography? This relates back to my earlier comments on the representativeness of this site.

*Ranai island is flat overall and the airport release site sits at 2 meters above mean sea level (AMSL). The Singapore city-state is located in the island lowlands at under 200 meters AMSL with the release site at 16 meters AMSL. Nevertheless, the topography north of Singapore on the Malay Peninsula encompasses the Titiwangsa Mountains, which peak at over 2,000 meters AMSL at Mount Korbu. Like Singapore, the release site at Puerto Princesa is also at 16 meters AMSL and is surrounded by mountainous terrain on the island of Palawan. Here the elevation also surpasses 2,000 meters AMSL at Mount Mantalingajan. While both Singapore and Puerto Princesa reside in the island lowlands, the sites are still influenced by the irregular mountainous and jagged coastal terrain. Ranai will still be influenced by the larger islands in the area, but is further removed from high topography.*

Page 9, line 26: What is meant by a "portion region of the southwest monsoon"?

*This is a typo. It has been edited to read "a portion of the southwest monsoon."*

Fig. 9. Is perturbation temperature being plotted here? If so, please specify in figure caption. What are the differences between the black and white spaces? Would it be possible to linearly interpolate in time the missing data, at least for reasonably short time gaps, shown in the right-hand panels? This would make the structures referred to in the text refers to easier to identify. In the last sentence of the caption are colors red and blue reversed.

*Perturbation temperature is being plotted and the caption has been updated, along with the color description for the MJO. The figure has also been updated to remove black spaces and has been interpolated for short time scales (< 5 days of missing data in a row).*

Is any ENSO time-scale signal detected in the RCs?

*Currently no ENSO signal is apparent in the RCs, although a future bin-analysis may reveal one. The ENSO signal may also lie in a higher order RC that the researchers did not feel comfortable applying physical meaning to. Lastly, the signal may be embedded in variability already captured by the current RCs.*

Page 10, lines 9-10: Does RC1 of temperature shown in Fig. 10 reflect differences in mean tropopause height of these sites with Singapore have the highest mean height and Ranai the lowest? If so, would it be possible to list the mean tropopause heights of these three sites?

*Both RC1 and RC5 represent tropopause height. Looking at only the most recent full year of data (2016) the mean tropopause height reported for the soundings are Ranai: 91.83 hPa; Puerto Princessa; 91.38 hPa, and Singapore: 90.84 hPa. This pressure level is calculated via the World Meteorological Organization's definition of the tropopause: "The lowest level at which the lapse rate decreases to 2 °C/km or less, provided that the average lapse rate between this level and all higher levels within 2 km does not exceed 2 °C/km." While the results follow the relative order of the RCs, the total range between the sites is only 1°C and may not be significant. Small changes in quality control procedure or number of data points could easily alter the mean values.*

*Furthermore, the calculation of tropopause height is somewhat arbitrary and difficult to interpret. For the purpose of this study, it's appropriate to envision the tropopause as a transition layer, rather than an individual point. Because the PCA algorithm was applied to the vertical column, it captures the variability due to this transition from the troposphere into the stratosphere.*

In Fig. 1 instead of using some random MTSAT image it would be more instructive if the long-term mean rainfall map for this region were shown. For example, this map could be based on MERRA analyses or preferably TRMM or GPCP observations which are easily accessible. In this manner the reader could see how the sites chosen were located with respect to regions of major convection.

*A TRMM seasonal climatology plot has replaced the MTSAT image as Figure 1 (Supplementary Figure 4)*

Page 3, line 27: Please state when RH is first mention in the paper whether it was computed with respect to ice or water at temperatures less than 0C.

*RH was calculated with respect to water for the entire vertical column, including at temperatures less than 0 °C. Switching the calculation from water to ice would produce a discontinuity, which the PCA algorithm does not handle well.*

**Technical corrections:**

Page 1, line 9: continent misspelled.
Page 1, line 13: I suggest changing "sounding release sites" to "upper-air sounding sites" here and other places where this phrase is used.
Page 5, line 29: decomposition misspelled.
Page 4, line 9: Suggest rewording, "... the database was subjectively analyzed in the following fashion."
In Table 1 title and caption change "Control Flags" to "Checks". Also suggest changing "checkpoint in" to "pass at". Finally suggest changing last sentence in caption to: "This is not a comprehensive list as additional subjective QC was applied."
In a few of the figures (Figs. 6, 9 and 10) where lines overlap, it's difficult to distinguish between black and dark blue lines. You might try using cyan instead of dark blue to make the lines more distinguishable.

*The above technical suggestions and corrections have all been addressed and updated in the paper.*

**General comments:**

The papers examines temporal and spatial modes of atmospheric variability over the Maritime Continent (MC) by applying principal component analysis to upper-air data from three "representative" sounding sites. With this approach they conclude that the fundamental modes of spatial and temporal variability of this region can be captured with small set of coherent structures. The manuscript is reasonable well written and organized but could you use some additional clarification in several places as noted in comments below.

In addition, I was left wondering if the three sites examined, all of which border or are within the South China Sea, are truly representative of the entire the MC region which spans over 60 degrees of longitude and contains over 17,000 islands. It's unclear to what degree the RC analyses of these three sites capture the major modes of variability in this large MC region? What measures did the authors take to conclusively demonstrate this point? Were other sites examined, say sites bordering the Java Sea or coastal sites in Borneo or Papau New Guinea where topographic effects and the diurnal cycle dominate substantially more than the three sites examined here (see Peatman et al. 2014, QJRMS) or ENSO effects might be more prominent? Short of such efforts, the authors may need to temper their conclusions somewhat to reflect the more limited scope of their analyses.

*The limiting factor in expanding this work across other MC sites is the need for a reliable radiosonde release record. While many MC sites have a decent record of radiosonde observations, the availability of moisture observations above 250 hPa is scarce. The transition from the troposphere into the stratosphere is an important source of variability and should be included. Ranai, Singapore, and Puerto Princesa all report humidity beyond 250 hPa for a significant portion of their release history. Only a handful of other MC sites can be included in the analysis with this requirement, and generally the recording history of sites that report above 250 hPa is short, or the location is close to one of the three original sounding sites in the South China Sea. A statement has been added that most of the focus of this study will be the South China Sea region, but that the methodology can be applied elsewhere.*

*Nevertheless, the authors tested three additional sites at Koror, Palau (2008-2016), Sorong, Indonesia (2014 through 2016), and Cilicap, Indonesia (2014 through 2016). The number of observations for the additional sites can be found in Supplementary Table 1. Many of the structure functions between the sites are identical (Supplementary Figure 5), even at Cilicap and Sorong, which have much shorter release histories. Koror has the most robust release history and is similar to Ranai throughout. It is likely that the major differences between Ranai, Sorong, and Cilicap exist because the full spectrum of of short-term climate variability is not being captured in the sparse record history. Although, the fact that the PCA algorithm can identify signals at Sorong and Cilicap speaks to its robustness as a statistical method. Furthermore, it also demonstrates that a majority of the variability in the thermodynamic system is the result of predominant tropical meteorology and climate, specifically tropical waves and the monsoon.*

While it might be beyond the scope of this study, it would be helpful to put the variability of these MC sites in context by showing how they differ from sites in the Indian Ocean and West Pacific (e.g, Gan and Manus) where MJOs often typically

initiate and dissipate and even a midlatitude continental site which should show dramatically different structures. Another natural extension of this work would be a PC analysis of rainfall at each site to better understand the relationship of the RCs presented in this paper to convection.

5 *While outside the scope of this project, the researchers have started looking at sites outside the Maritime Continent, both in the tropics and the mid-latitudes. This will be the subject of future work and will include a bin analysis, where observations will be placed in categories based on the background environment such as MJO phase, monsoon onset, and ENSO index.*

**Additional Citations**

*Ciesielski, P. E., and R. H. Johnson, 2006: Contrasting Characteristics of Convection over the Northern and Southern South China Sea during SCSMEX. Mon. Wea. Rev., 134, 1041-1062, doi: http://dx.doi.org/10.1175/MWR3113.1.*

*Houze, R. A., S. G. Geotis, F. D. Marks, and A. K. West, 1981: Winter Monsoon Convection in the Vicinity of North Borneo.*
15 *Part I: Structure and Time Variation of the Clouds and Precipitation. Mon. Wea. Rev., 109, 1595-1614, doi: http://dx.doi.org/10.1175/1520-0493(1981)109<1595:WMCITV>2.0.CO;2*

*Nuret, M., J.-P. Lafore, F. Guichard, J.-L. Redelsperger, O. Bock, A. Agusti-Panareda, and J.-B. N'Gamini, 2008: Correction of humidity bias for Vaisala RS80-A sondes during the AMMA 2006 observing period. J. Atmos. Oceanic Technol., 25, 2152–*
20 *2158, doi:10.1175/2008JTECHA1103.1.*

*Wang, J., and L. Zhang, 2008: Systematic errors in global radiosonde precipitable water data from comparisons with ground-based GPS measurements. J. Climate, 21, 2218–2238, doi:10.1175/2007JCLI1944.1.*

25 *Wang, J., L. Zhang, A. Dai, F. Immler, M. Sommer, and H. Vömel, 2013: Radiation dry bias correction of Vaisala RS92 humidity data and its impact on historical radiosonde data. J. Atmos. Oceanic Technol., 30, 197–214, doi:10.1175/JTECH-D-12-00113.1.*

**Supplementary Figures**

[Figure]

*Supplementary Figure 1) Mean bias test performed on the rotated relative humidity components. An arbitrary mean humidity bias was introduced to the entire Ranai data set (left) and run through the PCA algorithm. On the right, the rotated RH components from the original Ranai data set (dark colors) plotted together with the rotated RH components from the biased data set (dashed light colors). There are no distinguishable differences between the original data and the data with an introduced mean bias.*

[Figure]

*Supplementary Figure 2) Comparison of rotated relative humidity components at the Singapore site (left) before and after the sonde type was switched on 21-Dec-2011. The data set was split in two on the day the sonde type was changed from the Graw sonde (light dashed lines) to the VRS92 sonde (light dotted lines). The PCA algorithm was then run on the two split data sets and the full period (dark continuous lines). There are slight variations at levels above 200-hPa and near the 700-hPa*

*level. This process was repeated for the Koror site (right), which also shows signs of slight variations in the upper levels when the data set is split in two.*

[Figure]

*Supplementary Figure 3) Rotated component weight time series. RC-weights (black dots) are accompanied by a best fit line (red) to highlight their cyclic nature. The best fit line was calculated with a weighted linear least squares method combined with a 2nd degree polynomial local regression with a span no larger than 5%. In almost all of the RCs, the dominant signal is the seasonal / monsoon cycle.*

[Figure]

*Supplementary Figure 4) TRMM 1998-2016 average seasonal rainfall climatology with the release site locations listed for reference.*

| Number of Soundings (2008-2016) | | | | |
|---|---|---|---|---|
| Variable | Ranai | Koror | Sorong | Cilicap |
| Temperature | 3429 | 5304 | 624 | 566 |
| Relative Humidity | 1401 | 5259 | 623 | 561 |
| U and V Wind | 3380 | 5237 | 625 | 548 |

*Supplementary Table 1) Total number of quality controlled radiosonde observations for each of the additional upper-air sounding sites.*

[Figure]

*Supplementary Figure 5) Additional release site comparison including Koror, Sorong, and Cilicap. Koror has the most robust release history and the RCs closely match the Ranai RCs. Sorong and Cilicap have much shorter release records and deviate more from the Ranai RCs.*

**Response to Reviewer 2 Comments**

*To begin, the authors would like to thank the reviewer for their time, attention to detail, and thoughtful insights on the paper and research. Each comment will be addressed point by point.*

**General Comments**

However, I am not really sure what new it brings to the table in addition to the usual procedure of generating T/RH composites about high rain events, and I think the paper oversells how useful this technique is likely to be in the future. For example, from work by Kiladis and others, we already know a lot about the temperature and wind anomalies associated with different types of convectively coupled waves in the tropics. This type of procedure, in which we project anomalies on to the types of physically known propagating
10  3D convectively coupled waves seems more insightful than the 1D PCA done here, in which there is no attempt to physically separate any of the myriad of influences on a particular profile.

*Looking at a 1D-representation is useful because interpretation is uncluttered, which can illuminate confounding signals and relationships. Attempting to model 3D meteorological processes is extremely difficult and the results are not always straightforward*
15  *due to model error and fundamental gaps in our process-level knowledge. This project is an attempt to take a step backward and ask / answer fundamental questions about the variability of thermodynamics in the MC region: Do we know which layers of the atmosphere are the most variable? How can we quantify this variability? How do perturbations to one of these layers affect adjacent and non-adjacent layers? Which thermodynamic states are most often observed? Analysis of the response of various modes of tropical convection to changes in the environment is only possible after we sufficiently understand the answers to these elementary*
20  *questions (or represent them quantitatively).*

*This relatively simple approach uses PCA, which is not to be confused with a factor analysis. The two methods are not synonymous techniques and have unique motivations for their respective applications. PCA seeks to identify where variability occurs and remove data redundancies to reduce dimensionality. PCA has no prior assumptions about the underlying causes of the identified variability,*
25  *which can be both a strength and a weakness. Conversely, factor analysis aims to attribute the known system variability to a set of predefined processes and influences. A factor analysis performed on variable layers identified from the PCA would be an interesting project, but it is outside the scope of this study.*

*We employed the PCA approach because it is easily implemented, the outcome is straightforward and intuitive, and it is based on*
30  *observations rather than modeled output. The results can be used in several ways from both a data analysis point of view, such as expanding into a factor analysis, or from a modeling perspective. The observational RCs can be used to test whether models have accurately represented variability in the system or as initialization / boundary conditions in ensemble-based modeling techniques.*

**Specific Comments**
35  (1)  Section 4.1. The justification for the physical interpretations to which the various RC's are assigned is often unclear. For example, "shallow convective heating is represented in RC4".

*Interpretation of the PCs / RCs is admittedly subjective. When the physical attribution was not obvious, we relied on Figure 9 as a starting point. A motivation behind this project was to rely on a statistical framework to identify variable layers, rather than*
40  *preconceived ideas. Fortunately, by comparing the RCs to the time series in Figure 9 we see that oftentimes our intuition about which layers might be variable (e.g. that the melting layer, top of boundary layer, and the tropopause transition would all be good candidates for regions with significant thermodynamic variability) are also identified by the PCA. While to first order, our eyes could distinguish these layers from Figure 9, we now have a robust mathematical representation of their location and magnitude and an understanding of how variability in one layer correlates with variability in surrounding levels. The last point is the most problematic*
45  *when approaching this problem from a non-statistical framework. We could infer the relationships or attempt to model them, but the signal is already present in existing observations and can be captured through the PCA algorithm.*

*Depending on the application of the PCs / RCs, precise physical interpretation may or may not be necessary. From a modeling perspective, whether the interpretation exists or not the information on variability is retained and can be used to initialize or compare*
50  *simulations. Conversely, from a data analysis perspective, it is important that the interpretation is correct. Overall, the physical attribution has to answer two questions "where would variability in this layer originate from on a basic level, and on what time scales is it acting or being modified?"*

55  But do we really know what kinds of temperature anomalies are likely to be associated with shallow convective heating? For example, if shallow convective clouds occur more frequently (e.g. are triggered) by the moistening and cooling associated with low level upward motion (likely, especially in the vicinity of deep convection), then perhaps shallow clouds are correlated with low level cold anomalies, and any positive correlation between shallow convection with positive RH should not be interpreted as a consequence of detrainment moistening, but some external dynamically imposed influence.

*The variability associated with what we infer to be the action of shallow clouds (RH5 and T4) is captured exactly as described here. The shallow convective heating peaks at 700 hPa and has an associated cooling below 850 hPa. As stated, we can infer that the shallow clouds are correlated with low-level cold anomalies. This is also apparent in Figure 9, where warm temperature anomalies near 700 hPa sit atop cold anomalies, and vice versa. This could be the result of cooling via ascent, evaporative cooling, and / or radiative effects.*

For example, even the net effect of precipitating shallow convection on the RH of a particular level is unclear. It is a residual of the drying associated with induced descent, moistening from detrainment and evaporative moistening, and then a slower dynamical response driven by the geopotential anomalies associated with the convective heating.

*In most cases, this method does not allow ascription of the individual forcing or combination of environmental factors that go on to produce the observed variability. This sort of attribution would require an in-depth synoptic and mesoscale analysis for every observation in the dataset. While the source may be dynamic, the result is still going to manifest in the thermodynamic signature in predictable ways. Naturally, many different dynamical and / or microphysical processes could go on to produce the same thermodynamic signature, but this sort of factor analysis will have to be part of a future study.*

*Furthermore, in this more basic framework we can only focus on correlations between layers for one variable at a time. At this point the method is limited in that we cannot relate the thermodynamic variables to each other. Correlations between the thermodynamic variables will also have to be explored in future work.*

More generally, causality between T, RH, u, v anomalies in the background atmosphere and convective clouds always goes both ways. There can't be a simple one to one relationships between certain types of T/RH anomalies and certain cloud types or heating profiles, as implied here. (Otherwise it seems to me that convectively coupled waves in the tropics could not exist.)

*The tropical wave cycle is ideal for a PC analysis. The PCA results in both a positive and a negative mode for each PC (Figure 7), which would be required to describe convectively coupled waves. For simplicity, only the positive mode was displayed in Figures 8 and 10, although the opposite mode is also valid. Thus, for a convective disturbance (say a positive RH mode), there would be an associated negative RH mode preceding and following the wave.*

(2) Similarly, sometimes the RC's for U and V are assigned physical interpretations and again the justification is unclear. E.g. "The overwhelmingly dominant signal in the V-component of wind is the seasonal monsoon. The MC monsoon is characterized by a complete reversal ...". I guess it is not clear to me here what exactly is meant by "monsoon" in a region of such complicated topography, or why it must have these impacts on U and V. For example, the three radiosonde locations are at quite different locations in the Marine Continent, so the dynamical signature of the monsoon must vary between locations, but the RC's of the three locations are the same almost (except for ordering).

This is noted in point 4 of the conclusion. Should the "monsoon" have the same dynamical signature in all three locations? Perhaps give some explanation of what is really meant by "monsoon". It seems that the authors have simply defined a particular RC as a monsoon signature, and then remarked that this RC is the same at all three locations, and then say the monsoonal signature is the same at all three stations. Everything proceeds from the initial categorization. But is this really more than a semantic game? Do you really know for certain what types of large scale dynamical motions are associated with a particular RC? How would you prove this? I realize there is some discussion of this in lines 13-14 of Section 4.1, but this wasn't fully convincing to me.

*The temporal signature of variability is generally lost during PCA. However, some structure of this remains in the RC-weight time series (Supplementary Figure 1). Looking at the figure, for almost all of the RCs, the seasonal monsoon is the dominant cyclical signal. The three sites in the study are all in the South China Sea, so it is possible that at different sites the monsoon signal would be stronger, weaker, or non-existent. The sites may feel the influence of the monsoon at different times, but because the RCs in themselves are non-temporal, the physical wind reversal is all that can be captured.*

*Because this analysis is on a short-climatology timescale, individual dynamical motions are not the desired result. The much higher resolution time scales would be a perfect project for modelers, or would require a much more in-depth data and meteorological examination than is possible here. This is only a first step in a much larger and more complicated problem. But looking at variability in a lower dimensional subspace allows researchers to ignore system noise (the individual dynamic motions) and categorize specific modes of variability.*

(3) Figure 9. I found this hard to interpret. Especially there was so much variability in the top 4 panels, that the features discussed in the text were not clear to me.

*The figure has been updated to remove black spaces and has been interpolated for short time scales (< 5 days of missing data points in a row). Signals are now more apparent in the variability and matching the RCs to the panels in Figure 9 is now more intuitive.*

***Supplementary Figures***

[Figure]

*Supplementary Figure 1) Rotated component weight time series. RC-weights (black dots) are accompanied by a best fit line (red) to highlight their cyclic nature. The best fit line was calculated with a weighted linear least squares method combined with a 2nd degree polynomial local regression with a span no larger than 5%. In almost all of the RCs, the dominant signal is the seasonal / monsoon cycle.*

**Revised manuscript with changes tracked follows**

[revised manuscript text omitted]